# CD109 Is a Critical Determinant of EGFR Expression and Signaling, and Tumorigenicity in Squamous Cell Carcinoma Cells

**DOI:** 10.3390/cancers14153672

**Published:** 2022-07-28

**Authors:** Shufeng Zhou, Amani Hassan, Tenzin Kungyal, Sebastien Tabariès, José Luis Ramírez García Luna, Peter M. Siegel, Anie Philip

**Affiliations:** 1Departments of Surgery and Medicine, Division of Plastic Surgery, the Research Institute of the McGill University Health Center, McGill University, Montreal, QC H3G 1A4, Canada; shufeng.zhou@mail.mcgill.ca (S.Z.); amani.hassan@mail.mcgill.ca (A.H.); tenzin.tenzinkungyal@mail.mcgill.ca (T.K.); jose.ramirezgarcialuna@mail.mcgill.ca (J.L.R.G.L.); 2The Goodman Cancer Institute, Faculty of Medicine, McGill University, Montreal, QC H3A 1A3, Canada; sebastien.tabaries@mcgill.ca (S.T.); peter.siegel@mcgill.ca (P.M.S.)

**Keywords:** CD109, EGFR, squamous cell carcinoma, signaling, tumorigenesis

## Abstract

**Simple Summary:**

Squamous cell carcinoma is a type of cancer resulting from cancer of the squamous cells that line some organs of the body such as the skin, lungs, and vulva. It is one of the leading causes of cancer-related deaths worldwide. In the current study, we show that a protein called CD109 plays an important role in the progression of this type of cancer. We also show that CD109 acts by associating with another protein called the epidermal growth factor receptor (EGFR), stabilizing its levels and promoting its action, leading to increased progression of squamous cell carcinoma. Together, these findings highlight a potential clinical utility for targeting CD109 in squamous cell carcinoma.

**Abstract:**

(1) Background: Squamous cell carcinoma (SCC) is one of the leading causes of cancer-related deaths worldwide. CD109 is overexpressed in many cancers including SCC. Although a pro-tumorigenic role for CD109 has been shown in non-SCC cancers, and in one type of SCC, the mechanisms and signaling pathways reported are discrepant. (2) Methods: The CD109-EGFR interaction and CD109-mediated regulation of EGFR expression, signaling, and stemness were studied using microarray, immunoblot, immunoprecipitation, qPCR, immunofluorescence, and/or spheroid formation assays. The role of CD109 in tumor progression and metastasis was studied using xenograft tumor growth and metastatic models. (3) Results: We establish the in vivo tumorigenicity of CD109 in vulvar SCC cells and demonstrate that CD109 is an essential regulator of EGFR expression at the mRNA and protein levels and of EGFR/AKT signaling in vulvar and hypopharyngeal SCC cells. Furthermore, we show that the mechanism involves EGFR-CD109 heteromerization and colocalization, leading to the stabilization of EGFR levels. Additionally, we demonstrate that the maintenance of epithelial morphology and in vitro tumorigenicity of SCC cells require CD109 localization to the cell surface. (4) Conclusions: Our study identifies an essential role for CD109 in vulvar SCC progression. We demonstrate that CD109 regulates SCC cellular stemness and epithelial morphology via a cell-surface CD109-EGFR interaction, stabilization of EGFR levels and EGFR/AKT signaling.

## 1. Introduction

Squamous cell carcinoma (SCC), especially head and neck SCC, is one of the leading cause of cancer-related death worldwide [1]. Despite surgical approaches to control early-stage tumor growth, patients often remain disfigured and have poor prognoses. So far, no effective targeted regimens for advanced SCC in patients have been identified [2]. To establish SCC prevention and treatment strategies successfully, a better understanding of the molecular events underlying SCC tumorigenesis is necessary.

CD109, a member of the α2 macroglobulin (α2M)/C3 family, is a glycosyl-phosphatidylinositol (GPI)-anchor cell surface glycoprotein highly expressed in many cancers, including SCCs [3,4]. Previous work from our laboratory provided the first evidence that CD109 is a component of the transforming growth factor-β (TGF-β) receptor system, acting as a TGF-β coreceptor and negative regulator of TGF-β signaling. Thus, CD109 strongly inhibits TGF-β-induced extracellular matrix (ECM) synthesis and other responses such as the epithelial-to-mesenchymal (EMT) [5,6,7,8,9,10,11]. Moreover, we previously showed that the mechanism by which CD109 inhibits TGF-β signaling involves CD109 forming a heteromeric complex with the types I and II TGF-β signaling receptors and promoting TGF-β receptor compartmentalization and internalization via the caveolae leading to TGF-β receptor degradation in a ligand-dependent manner [5,6,7,8,9,10,11]. In addition, we and others reported that CD109 is endogenously released from the cell surface, and the released/soluble form is able to bind TGF-β and inhibit TGF-β signaling and responses [7,8,11,12,13].

The expression of CD109 is dysregulated in many cancers including SCC, and high levels of CD109 are frequently detected in premalignant lesions of the oral cavity, which are associated with a significantly higher risk of progressing toward an overt SCC [14,15,16,17]. We previously demonstrated that CD109 functions as a gatekeeper of the epithelial phenotype by suppressing TGF-β-induced EMT in SCC cells and that loss of CD109 promotes a transition to the mesenchymal phenotype and increases the motility of SCC cells, suggesting a potential anti-tumorigenic role [18]. However, increasing evidence suggests that CD109 may play a pro-metastatic role in several types of non-SCC cancers. For example, it was recently reported that CD109 drives metastasis in lung adenocarcinoma by modulating Jak/Stat3 signaling [19]. In SK-MG1 glioblastoma cells, while CD109 attenuates TGF-β signaling, it also enhances epidermal growth factor receptor (EGFR) signaling [20]. A recent study showed that CD109 mediates tumorigenicity in cervical SCC and that it involves regulation of EGFR/STAT3 signaling [21]. While the above studies made great strides in underscoring a role of CD109 in cancer progression, they yielded discrepant results on the pathways involved. Defining the precise mechanisms by which CD109 regulates major oncogenic pathways in other types of SCC is crucial for the identification of molecular targets for therapeutic intervention.

EGFR signaling plays a central role in cell proliferation and survival [22,23], and aberrant activation of EGFR is known to be a major oncogenic pathway for progression and metastasis of many cancers [24,25,26,27] including SCC [28,29]. It is important to establish not only that CD109 plays a causal role in SCC tumor growth or metastasis but also to elucidate the molecular mechanisms by which CD109 regulates the oncogenic pathways such as EGFR signaling.

In the present study, we demonstrate that the loss of CD109 reduces stemness and tumorigenicity in vitro and abrogates tumor initiation and the metastatic ability of vulvar SCC cells in vivo, indicating an oncogenic role for CD109 in vulvar SCC. Furthermore, we show that CD109 is an essential regulator of EGFR expression at the mRNA and protein levels and of EGFR/AKT signaling in vulvar and hypopharyngeal SCC cells. In addition, we show that the mechanism of CD109 action in SCC cells involves EGFR-CD109 heteromerization and colocalization, leading to the stabilization of EGFR levels. We also demonstrate that the maintenance of epithelial morphology and in vitro tumorigenicity of SCC cells require CD109 localization to the cell surface.

## 2. Materials and Methods

### 2.1. Cell Culture

The human squamous carcinoma A431 (CRL1555) cell line (ATCC, Manassas, Virginia, USA) derived from a vulvar epidermoid carcinoma of an 85-year-old female) was maintained in Dulbecco’s modified Eagle’s medium (DMEM; Gibco Thermo Fisher Scientific, Waltham, MA, USA; 11995-065) supplemented with 10% fetal bovine serum (Gibco Thermo Fisher Scientific, Waltham, MA, USA; 12483-020). The human head and neck squamous carcinoma FaDu (HTB-43) cell line (ATCC, Manassas, VR, USA) derived from a hypopharyngeal SCC tumor of a 56-year-old male patient) was maintained in Eagles’s minimal Essential medium (EMEM; Gibco Thermo Fisher Scientific, Waltham, MA, USA; 670086) supplemented with 10% fetal bovine serum (Gibco Thermo Fisher Scientific, Waltham, MA, USA; 12483-020).

### 2.2. Generation of Knockout Cell Line with CRISPR/Cas9

The generation of stable CD109 KO cell lines was achieved as we previously described [18]. Briefly, the complementary oligo-nucleotides for guide RNAs (gRNAs) were cloned into the pX458 CRISPR /Cas9-GFP vector (Addgene, Watertown, MA, USA). Vulvar SCC (A431) cells were transfected with either pX458/gRNA #1 or pX458/gRNA #2 using lipofectamine 2000, according to the manufacturer’s instructions. After transfection, and expansion of GFP positive cells, they were stained for CD109 and sorted as a single CD109 negative cell per well in a 96-well plate. The single CD109 negative cell was used to generate each CD109 KO single cell clone (KO26, KO32, and KO180). The control cells consisted of A431 cells which went through the same cloning and GFP sorting procedure as the experimental groups (GFP^+^/CD109^−^ cells) but did not have CD109 knocked out (GFP^+^/CD109^+^ cells), as they retained CD109 expression and epithelial traits, and hereon these A431CD109WT will be referred to as A431 control cells in the text and figures. We previously showed that these A431 control cells display similar characteristics as parental A431 cells, as we found no differences in any of the end points analyzed in studies of A431-parental cells (sorted for CD109 high vs. CD109 low) and the A431 control clone (and CD109 knock out clones) [18].

### 2.3. siRNA Transfection

siRNAs directed against CD109 (Invitrogen Thermo Fisher Scientific, Waltham, MA, USA #4392420) with the sense sequence 5′-GAUCUAUCCAAAAUCAAGAtt-3′ and antisense sequence 5′- UCUUGAUUUUGGAUAGAUCtt-3′ or control scrambled siRNAs (Invitrogen Thermo Fisher Scientific, Waltham, MA, USA #AM4611) were transfected into FaDu cells using the Lipofectamine RNAiMAX transfection reagent (Invitrogen Thermo Fisher Scientific, Waltham, MA, USA) according to the manufacturer’s instructions. At 48 h post-transfection, cells were serum starved for 24 h (to diminish growth factors/cytokines). The FaDu cells with CD109 knock down were used for immunoblotting, Co-IP, and immunofluorescence analysis.

### 2.4. Toluidine Blue Staining

Cells were fixed in 4% PFA for 15 min, washed 3 times with PBS, followed by 5 min of staining in 0.1% toluidine blue, quickly dipped in dH_2_O, and let to air dry.

### 2.5. In Vivo Animal Studies

NOD-SCID or Beige-SCID male mice (6–8 weeks old) were purchased from Charles River. All animals were housed in a specific pathogen-free facility and had ad libitum access to water and food and were maintained on a 12-h light/12-h dark cycle. All studies were approved by the Animal Resource Centre at McGill University and complied with guidelines set by the Canadian Council of Animal Care.

### 2.6. In Vivo Tumor Xenograft Formation

For these studies, we used 3 different CD109 KO A431 clones (GFP^+^/CD109^−^, experimental groups). The control group consisted of A431 cells which went through the same cloning procedure but retained CD109 expression, representing GFP^+^/CD109^+^ cells. There were 24 mice per experiment with 6 mice/group. Each mouse represents an experimental unit. Cells (>95% viable) in the PBS/Matrigel mixture (1:1 volume) (Corning Matrigel Growth Factor Reduced; 354230) (1 × 10^6^/mouse) were injected subcutaneously into right flanks of 5-week NOD/SCID or Beige/SCID male mice. Mice were maintained for 5 weeks and inspected for tumor appearance, by observation and palpation, and tumor growth was measured weekly using a caliper. Tumor volume was determined using the standard formula: L × W^2^ × 0.52, where L and W are the longest and shortest diameters, respectively. At the end of the experiment, mice were anesthetized with isoflurane and sacrificed by CO_2_ asphyxiation followed by cervical dislocation. The presence of each tumor nodule was confirmed by necropsy.

### 2.7. Tail Vein (Metastasis) Assay

Twenty Beige/SCID mice were randomly divided into five groups: Control group (injected with only 0.1 mL PBS), A431 control clone, CD109-KO clone 26, clone 32, and clone 180. The injections of cells involved resuspending 1 × 10^6^ cells (A431 control clone or one of the three CD109-KO clone cells, namely KO26, KO32, and KO180) in 0.1 mL PBS and directly injecting into the lateral tail vein. After 6 weeks, mice were sacrificed as described above, and lungs were removed at necropsy to be assayed for the presence of metastatic lesions and were sectioned for H&E staining and IHC staining. Metastatic burden in the lungs was quantified from four H&E-stained step sections (40 mm/step). The metastatic lesion area/lung area was quantified via Imagescope software (Aperio).

### 2.8. Immunohistochemistry (IHC) Analysis

Lungs were collected from mice injected with the control (PBS), or control A431 or a knock out clone (KO180, KO32, or KO26), and lung tissue sections were analyzed by IHC. Incubations with the primary antibodies were conducted overnight at 4 °C for: mouse monoclonal anti-CD109 (Santa Cruz, Santa Cruz, CA, USA; sc271085), EGFR antibody (Cell Signaling, Danvers, MA, USA, # 4267S). The sections were washed and incubated with secondary antibodies (Advanced HRP Link, DakoCytomation, K0690, Denmark) followed by the polymer detection system (Advanced HRP Link, DakoCytomation). Reactions were developed with a solution containing 0.6 mg/mL of 3,3′-diaminobenzidine tetrahydrochloride (DAB, Sigma, St Louis, MO, USA) and 0.01% H_2_O_2_ and then counter-stained with Mayer’s hematoxylin. Positive controls (a tissue known to contain the antigen under study) were included in all reactions in accordance with the manufacturer’s protocols.

The negative control consisted of omitting the primary antibody and incubating slides with PBS and replacing the primary antibody with normal serum. The IHC was performed for at least 3 mice for A431, KO180, KO32, and KO26.

### 2.9. Immunofluorescence Staining

Cells plated on coverslips were fixed in 4% paraformaldehyde (*w*/*v*) for 15 min, and permeabilized in PBS/0.3% Triton X-100 for another 15 min. Cells were then washed with PBS and blocked in 2% BSA for 1 h. Primary antibodies namely, anti-CD109 (C-9) (Santa Cruz Biotechnology, Santa Cruz, CA, USA; sc-271085), anti-EGFR (Cell Signaling, Danvers, MA, USA; mAb #2232), anti-Phospho-EGF Receptor (Tyr1068) (Cell Signaling, Danvers, MA, USA; mAb #3777), anti-Sox2 (cell Signaling Danvers, MA, USA; mAb #3108), anti-Oct4 (Santa Cruz, Santa Cruz, CA, USA; sc-23900), and anti-NANOG (Santa Cruz, Santa Cruz, CA, USA; sc-23900) antibodies were then added to cells at 1:300 dilution in 2% BSA and incubated overnight at 4°C. Cells were washed with PBS and labeled for 1 h with fluorophore-conjugated secondary antibodies (1:500 dilution) specifically, Alexa Fluor 594-goat anti-rabbit (Life Technology; A11037) and Alexa Fluor 488 goat-anti-mouse (Life Technology; A11029). Cells were washed with PBS, and mounted with Fluoroshield mounting medium with DAPI (Abcam; ab104139). Cells were visualized using an Olympus microscope IX71.

### 2.10. Western Blot Analysis

Cell lysates (containing 20 μg total protein) were analyzed by western blot with one of the following antibodies: mouse monoclonal anti-CD109 (BD Biosciences; 556039), anti-fibronectin (BD Biosciences; 610078), anti-N Cadherin (Abcam; ab18203), anti-E-Cadherin (Abcam; ab216783), anti-p44/42 Erk1/2 (Cell Signaling, Danvers, MA, USA; mAb #4695), anti-STAT3 (Abcam; ab5073), Anti-phospho-Stat3 (Tyr705) (Cell Signaling, Danvers, MA, USA; #9131S), Anti-Akt (Cell Signaling, Danvers, MA, USA; mAb #4685), Anti-phospho-Akt (Ser473) (Cell Signaling, Danvers, MA, USA; mAb #4060), anti-β-actin antibodies (Santa Cruz Biotechnology, Santa Cruz, CA, USA; sc-47778). Briefly, western blot analysis involved separating cell extracts by SDS-polyacrylamide gel electrophoresis (PAGE) and transferring to polyvinylidene difluoride membranes. Membranes were blocked with 5% bovine serum albumin/TBST blocking buffer at room temperature for 30 min and then incubated overnight at 4 °C with specific primary antibodies. The membranes were washed with TBST wash buffer followed by incubation with a horseradish peroxidase-conjugated secondary antibody at room temperature for 1 h. Bands were detected with an enhanced chemiluminescence (ECL) system (MilliporeSigma, Burlington, MA, USA; WBKLS0500).

### 2.11. Co-Immunoprecipitation Experiments

Co-immunoprecipitation (Co-IP) was conducted using a Thermo Scientific Pierce Co-IP kit (Thermo Fisher Scientific, Waltham, MA, USA; 26149) as per the manufacturer’s protocol. Briefly, the CD109 antibody (BD Bioscience, Franklin Lakes, NJ, USA; 556039) or EGFR (Cell Signaling, Danvers, MA, USA; 4267S) was incubated with the delivered resin and covalently coupled. The antibody-coupled resin was then washed and incubated with cells’ lysate from the control A431 or CD109 KO A431 cells overnight at 4 °C. The resin was washed, and the protein complexes bound to the antibody were eluted, and western blot analysis was performed as indicated above (the covalent attachment and non-reducing elution system used in this method yielded pure co-IP products, while underivatized agarose beads served as a negative control for nonspecific binding). For FaDu cells, Co-IP was conducted using rabbit polyclonal anti-EGFR (Cell Signaling, Danvers, MA, USA; 4267S), or mouse monoclonal anti-CD109 (BD Bioscience, Franklin Lakes, NJ, USA; 556039), and Protein G magnetic beads as per the manufacturer’s protocol (Biorad Hercules, San Francisco, CA, USA; 1614023) and were analyzed by Western blotting with the anti-CD109 or anti-EGFR antibody or isotype control IgG. Horseradish peroxidase-conjugated secondary antibodies (Cell Signaling, Danvers, MA, USA; 7076S) were used for the Western blot after both Co-IP procedures.

### 2.12. TGF-β, EGFR and AKT Inhibition Studies

The TGF-β inhibitor SB431542 (Selleckchem, Radnor, PA, USA; 1067), EGFR inhibitor AG1478 (Selleckchem, Radnor, PA, USA; S2728), and AKT inhibitor MK 2206 (Selleckchem, Radnor, PA, USA; S1078) were dissolved in dimethylsulfoxide (DMSO; Sigma-Aldrich, Burlington, MA, USA) and used at the concentrations indicated. DMSO was used at an identical volume as a control.

### 2.13. RNA Isolation and Quantitative Real-Time PCR

Total RNA was isolated from cultured cells using the RNeasy mini-Kit (QIAGEN, Redwood City, CA, USA; 74104). RNA pellets were resuspended in diethylpyrocarbonate (DEPC)-treated water, and the RNA concentration was determined using the NanoDrop ND-2000 spectrophotometer (Thermo Fisher Scientific, Waltham, MA, USA). Reverse transcription (RT) was conducted on 2 μg of RNA using SuperScript III First-Strand Synthesis (Life Technologies Thermo Fisher Scientific, Waltham, MA, USA). The cDNA generated was subjected to qPCR on a 7500 Fast Real-Time PCR system using software version 2.0.4 (Applied Biosystems Thermo Fisher Scientific, Waltham, MA, USA). Gene expression was assayed using the Power SYBR Green PCR Master Mix (Applied Biosystems Thermo Fisher Scientific, Waltham, MA, USA) according to the manufacturer’s instructions. The primer sets used are shown in Table 1.

### 2.14. Generation of CD109 KO A431 Cells That Express EGFR (CD109KO^EGFR^)

The 80% confluent CD109 KO cells were transfected with EGFR-GFP plasmid (Addgene Watertown, MA, USA; #32751) using the Lipofectamine 2000 reagent (Invitrogen Thermo Fisher Scientific, Waltham, MA, USA) in Opti-MEM (Invitrogen Thermo Fisher Scientific, Waltham, MA, USA) as per the manufacturer’s protocol. An empty backbone pEGFP-N1-FLAG plasmid (Addgene, Watertown, MA, USA; #60360) was used for mock transfection.

CD109 KO A431 cells and CD109 KO A431 cells that express EGFR (CD109-KO^EGFR^) were co-transfected with one of the following plasmids: GPI-anchored CD109 (CD109G), which encodes GPI-anchored CD109 endogenously expressed on the cell membrane, or its empty vector, pCMVSport6 (EVG); soluble CD109 (CD109 S), which encodes the ectodomain of CD109, or its empty vector, pcDNA3-Gateway vector (EVS), as described previously [18], using lipofectamine 2000 (Invitrogen Thermo Fisher Scientific, Waltham, MA, USA) according to the manufacturer’s instructions.

For EGF stimulation experiments, cells were seeded overnight in serum-free-medium and then stimulated with 100 ng/mL EGF for 0, 15 min, 30 min, 1 h, and 2 h. Cell lysates were subjected to Western blot.

### 2.15. Tumor Spheres Formation Assay

A total of 10,000 single cells were suspended in a six-well ultra-low attachment plate (VWR, Radnor, PA, USA; 29443-030) at cells/well in DMEM/F12 medium with 20 ng/mL hEGF, 20 ng/mL hbFGF, and 2% B-27 (Life Technologies Thermo Fisher Scientific, Waltham, MA, USA) at 37 °C in 5% CO_2_. The medium was changed once a week. After two weeks, individual spheres were counted under an inverted microscope at 40× magnification. The percentage of cells capable of forming spheres was calculated as: (number of spheres formed/number of cells plated) × 100.

### 2.16. Statistical Analysis

All quantitative data are presented as mean ± SD. Data were analyzed by a two-tailed Student’s *t*-test or one-way and two-way ANOVA with Bonferroni post hoc testing as indicated. For repeated measures analysis, a two-way ANOVA was conducted. A *p*-value < 0.05 was considered significant.

## 3. Results

### 3.1. The Loss of CD09 Diminishes the Tumorigenicity and Abrogates the Metastatic Ability of SCC Cells In Vivo

We previously reported that loss of CD109 is associated with loss of the epithelial phenotype and enhanced cell proliferation and migration in vulvar SCC (A431) cells in vitro in 2-dimensional cultures [18]. Because enhanced EMT traits are traditionally related to stemness and metastasis, we sought to determine the impact of CD109 deletion on tumorigenicity of A431 cells in vivo. We subcutaneously injected 1 × 10^6^ control A431 or KO A431 cells (clone 32) into six-week-old male NOD/SCID mice at two flank sites. The mice were monitored every day and euthanized at 40 days post-transplantation. No adverse events were observed in the control or experimental groups. Twenty days after the injections, the majority of sites injected with control A431 cells formed palpable tumors, whereas no tumors could be detected at any of the sites injected with CD109-KO A431 cells. This was an unexpected finding based on our previous in vitro results showing that CD109 inhibits EMT traits and cell migration [18].

By the end of the experiment (day 40), none of the mice injected with CD109KO cells (0/6) developed any tumors, whereas all mice injected with A431 cells formed tumors (6/6) (Figure 1A–D), indicating that the loss of CD109 eliminated xenograft tumor growth of A431 SCC cells in vivo. Moreover, mice that received CD109-KO cells remained tumor free for an additional 6 months (Appendix A). NOD/SCID mice still possess natural killer (NK) cells, which could have possibly contributed to the elimination of the cancer cells. To avoid any artifacts resulting from mice genetic background or clonal characteristics of the chosen clone (KO32), we injected three independent CD109-KO clones into SCID-Beige mice that lack NK cell activity. Consistent with the previous results, no palpable tumors developed in any of the recipients (0/12 mice) that received the CD109-KO cells, whereas tumors developed within 5 weeks in SCID-Beige mice (4/4) that were injected with control A431 cells (Figure 1E–H). The human origin of the tumors was confirmed with immunohistochemistry staining for human marker Ku80, and the tumors exhibited typical SCC traits (Figure 1I). In addition, a H&E staining from lung and liver sections revealed no infiltration of cancer cells into these organs (Figure 1J–K).

Next, we tested whether the loss of CD109 was able to suppress the metastatic ability of SCC cells by tail vein assay. The control A431 and three clones of CD109 KO A431 cells were injected into SCID-Beige mice through the tail vein, and the lungs of these hosts were harvested for analysis 6 weeks post injection. Consistent with the in vivo tumorigenesis results, we observed a striking reduction in the metastatic capacity of CD109-KO cells to colonize the lung, a frequent site for SCC cells’ metastasis in mice (Figure 2B–E). Although all the mice that received control A431 cells developed intensive metastasis in their lungs, mice that received CD109-KO cells developed either no or very few metastatic lesions, and the metastatic burden in these recipients was reduced by more than 90% compared to mice injected with A431 cells (Figure 2B–E). We also observed that the weight of the lungs from mice that were injected with A431 cells was markedly heavier than those from the mice that received CD09-KO SCC cells (Figure 2B,C). H&E staining of lung sections revealed that the percentage of the lung metastatic area in mice that received CD109-KO cells was less than 10% of that in mice that received A431 cells (Figure 2D,E). Similarly, Micro-CT analysis of these lung sections revealed that the areas of lung tissue occupied by lung metastases in recipients that received CD109-KO cells were significantly smaller compared to the metastatic burden produced by control A431 cells (Appendix A). These observations demonstrated that the loss of CD109 severely impaired the formation of lung metastases, and the loss of CD109 potently inhibited metastasis of SCC cells in vivo. Analysis of the expression of CD109 and EGFR using IHC in the lung sections of mice injected with PBS (control), A431 control, KO180, KO32, or KO26 showed that, as expected, the expression of CD109 and EGFR is significantly lower in the lung sections of mice injected with KO clones, as compared to those injected with A431 control cells (Figure 2F).

### 3.2. CD109 Is Required for Maintaining the Stemness of SCC Cells, and the Loss of CD109 Diminishes the Cancer Stem Cell Population In Vitro in SCC Cells

As the results above suggest that CD109 is protumorigenic in SCC cells, we next examined the mechanism by which CD109 may exert its pro-tumorigenic effect by determining whether CD109 regulates the stemness of SCC cells. We first measured the expression of the pluripotent markers SOX2, OCT4, and NANOG in CD109 KO and control A431 cells by Western blot. Surprisingly, we found that the expressions of these pluripotent markers were attenuated in three independent CD109-KO clones (KO26, KO32, and KO180) when compared to control A431 cells (Figure 3A). This was further confirmed by immunofluorescence microscopy showing that CD109-KO cells (KO32) exhibit markedly reduced expression of these proteins (Figure 3D–F). Consistent with the above findings obtained using control and CD109 KO A431 cells, we found that the expressions of the pluripotent markers were decreased in CD109 knockdown (KD) hypopharyngeal SCC (FaDu) cells when compared to control FaDu cells (Figure 3A,D–F).

We next evaluated CD44 expression by flow cytometry in both CD109 KO and control A431 cells, as CD44 is a cancer stem cell (CSC) marker that is involved in the regulation of CSC survival, self-renewal, and metastatic colonization [30,31]. Our results show that the CD44-positive cell population is significantly decreased in CD109-KO SCC cells compared to WT A431 cells (Figure 3B,C), suggesting that upon the loss of CD109, A431 cells also lose CD44 expression, which is consistent with the previous report that CD44 and CD109 show similar expression patterns [32].

We then used in vitro tumor spheroid formation assays to determine whether, and how, the loss of CD109 affects cancer stem cell populations in SCC cells. After 2 weeks, the CD109-KO cells exhibited not only significantly fewer numbers of spheroids (3 times less) but also markedly smaller sized proliferating viable spheroids as compared to the control A431 cells (Figure 3G). To determine whether these spheroids had the potential to expand in vitro, the spheroids were dissociated into single cell suspensions and passaged multiple times in a long-term sphere-forming assay. Single cells from A431 spheroids repeatedly formed spheroids for up to five subsequent passages when plated in the absence of an attachment spheroid growth condition. However, cells from CD109-KO spheroids completely lost their ability to form spheroids after only three passages (data not shown and Appendix A). This indicates that A431 cells contain a significantly higher population of CSCs compared to the CD109 KO cells, suggesting that the loss of CD109 significantly diminishes the spheroid forming capacity of SCC cells.

### 3.3. CD109 Associates and Co-Localizes with EGFR, and Loss of CD109 Diminishes EGFR Expression in SCC Cells

To unravel the molecular mechanism of the dramatic effect of the loss of CD109 on tumorgenicity in SCC cells, we investigated global gene expression changes elicited by the loss of CD109. Microarray analysis highlighted several potential pathways as effectors of CD109 function, including EGF signaling (Figure 4A). Notably, the EGFR mRNA level was markedly downregulated in the three CD109-KO clones relative to the control A431 cells (Figure 4A). This was further confirmed by both RT-PCR and Western blot, showing a dramatic reduction in EGFR expression at both the mRNA and protein levels in CD109-KO SCC cells relative to control A431 cells (Figure 4B,C). Accordingly, the phosphorylation of EGFR (P-EGFR) and AKT (P-AKT) was also markedly decreased in the three CD109-KO cell lines as compared to control A431 cells (Figure 4C). Similar results were obtained in CD109 KD FaDu cells as compared to control FaDu cells, as detected by Western blot and immunofluorescence analysis (Figure 4C,D).

Next, we examined whether CD109 is associated or is co-localized with EGFR using co-immunoprecipitation and immunofluorescence co-localization studies. Immunoprecipitation of CD109 using anti-CD109 antibodies resulted in the co-immunoprecipitation of EGFR in both control A431 and FaDu cells (Figure 4E). However, no EGFR could be detected when CD109-was immunoprecipitated from CD109-KO A431 or CD109-KD FaDu cell lysates (Figure 4E). In the reciprocal IP experiment, immunoprecipitation of EGFR consistently co-precipitated CD109 in control A431 and FaDu cell lysates but not in the CD109-KO A431 and CD109-KD FaDu cell lysates (Figure 4F). Thus, reciprocal co-IP experiments showed a robust interaction between CD109 and EGFR in control A431 and FaDu cells, whereas this interaction was undetectable in CD109-KO and CD109-KD FaDu cells (Figure 4E–G). This was further validated by the colocalization of CD109 with EGFR using immunofluorescence microscopy using both A431 and FaDu cells. Results shown in Figure 4H demonstrated that both CD109 (green) and EGFR (red) were present on the cell membrane and are colocalized in both the control A431 and FaDu cells but are undetectable in CD109-KO A431 and CD109-KD FaDu cells (Figure 4H), supporting the notion that CD109 interacts with EGFR in control A431 and FaDu cells. The results shown for A431 KO cells were obtained using KO 32 A431 cells (Figure 4H right panel). Similar results were observed using all three A431 CD109 KO clones.

### 3.4. The AKT Pathway, but Not the ERK1/2 and STAT3 Pathways, Is Inactivated in CD109 KO SCC Cells, While in Control SCC Cells, EGFR Kinase Inhibition Phenocopies AKT Inhibition in Inducing Cell Death and CD109KO-Like Mesenchymal State

To determine how the loss of CD109 affects EGFR downstream pathways, we analyzed several effectors downstream of the EGFR pathway in SCC cells. The loss of EGFR activity in the three CD109-KO SCC clones correlated with a reduction in phospho-AKT levels compared to control A431 cells, while total AKT remained unaffected. In contrast to the decrease in phospho-AKT levels, the levels of phospho-ERK and phospho-Stat3 expression were unaltered in CD109-KO cells, as compared to control A431 cells (Figure 5A). These results suggest that the AKT pathway is specifically inactive, while the ERK and Stat3 pathways remain active in the CD109-KO SCC cells. To investigate the role of ERK/AKT pathway activation as compared to the activation of other pathways in the loss of epithelial morphology observed in the CD109 KO cells further, we first treated the control and CD109-KO A431 cells with SB431452, a TGF-β pathway inhibitor. As shown in Figure 5B, treatment with 10 µM/mL SB431452 did not affect cell morphology nor cell growth in both A431 and CD109-KO A431 cells, suggesting that TGF-β pathway inhibition is not responsible for the biological effects observed (Figure 5B,C). In contrast, treatment of the cells with low concentrations (5 µM/L) of AG1478, an EGFR inhibitor induced pronounced cell death in control A431 cells, but not in CD109-KO A431 cells, confirming a loss of EGFR signaling in CD109 KO cells (Figure 5D). Similar results were obtained with the inhibition of the AKT pathway with low concentrations (5 µM/L) of an AKT inhibitor, MK2206 (Figure 5E). Moreover, phosphorylation of both EGFR and AKT in A431 cells was markedly blocked upon the treatment of EGFR or the AKT inhibitor but not upon the treatment of the TGF-β inhibitor, which is consistent with the observation of intensive cell death in A431 cells upon the treatment with EGFR or AKT inhibitors (Figure 5B–G). This is in agreement with the marked apoptosis observed with the EGFR [33] and AKT [34] inhibitors in other cell types. In addition, a complete loss of AKT phosphorylation is observed in CD109-KO cells irrespective of TGF-β, EGFR, or AKT inhibitor treatment (Figure 5C,F,G), confirming that neither EGFR nor AKT pathways were activated in CD109-KO cells. The decrease in EGFR phosphorylation with the AKT inhibitor in control A431 cells is likely due to a feedback loop involving EGFR and AKT, as was reported previously [35].

### 3.5. GPI-Anchored CD109 Is Required for Maintaining EGFR Expression, Levels of Phospho-EGFR, and Phospho-AKT, and the Epithelial Phenotype in CD109KO SCC Cells

To understand the molecular mechanism underlying the decrease in EGFR levels in CD109 KO cells better, we overexpressed EGFR in CD109KO cells (CD109KO^EGFR^) and found that overexpression of EGFR in CD109KO cells had no effect on CD109 expression at both mRNA and protein levels (Appendix A). Surprisingly, overexpression of EGFR did not alter the morphology of the CD109-KO cells, as both CD109 KO cells and CD109KO^EGFR^ showed a similar morphology (Appendix A, Figure 6A). Although the overexpression of EGFR led to a robust increase in EGFR mRNA expression in the CD109KO^EGFR^, the expression of EGFR at the protein level was modest relative to that of control A431 cells, as detected by Western blot (Appendix A).

We next assessed if rescuing CD109 expression in CD109-KO cells could restore membrane localization of EGFR and restore the epithelial morphology in CD109 KO cells. We transiently overexpressed either GPI-anchored CD109 (CD109-KO^CD109G^) or soluble CD109 (CD109-KO^CD109S^) into CD109-KO cells. Interestingly, the CD109 KO cells overexpressing GPI-anchored CD109 (CD109-KO^CD109G)^) showed a robust expression of CD109 and a partial gain of epithelial morphology, while the CD109 KO cells overexpressing soluble CD109 (CD109-KO^CD109S^) displayed no appreciable changes in morphology (Figure 6A). In line with this, the expression levels of mesenchymal proteins (EMT markers such as E-cadherin, fibronectin) in the CD109-KO^CD109G^ cells were intermediate between control A431 cells and CD109-KO cells (Figure 6A,B), while the CD109-KO^CD109S^ cells showed no changes in EMT marker expression. This is consistent with our previous findings where the expression of EMT markers in control A431 vs. KO clones were quantified [18]. Together, these findings suggest that transiently forcing the expression of CD109 in CD109-KO SCC cells is not sufficient to restore the epithelial phenotype or EGFR expression. We then transiently overexpressed EGFR together with either the GPI-anchored-CD109 (CD109KO^EGFR+CD109G^) or soluble CD109 (CD109KO^EGFR+CD109S^) and found that the overexpression of EGFR with GPI-anchored CD109, but not the soluble form of CD109, can restore the epithelial phenotype, as can be observed in CD109KO^EGFR+CD109G^ cells (Figure 6A). Accordingly, expression of epithelial marker E-cadherin was markedly increased, and the mesenchymal protein, fibronectin, was decreased in the CD109-KO^EGFR+CD109G^ compared to cells transfected with the empty vector or to other transfected control cells (Figure 6B,C). Importantly, the EGFR expression and the EGFR/AKT pathway were restored in the CD109-KO^EGFR+CD109G^ cells (Figure 6C lane 2 vs. lane 5), further demonstrating that localization of EGFR at the membrane rescues the EGFR/AKT pathway to levels comparable to that of control A431 cells (Figure 6D). Consistent with this, immunofluorescence microscopy revealed that no EGFR protein was detectable in cells transfected with EGFR alone (CD109-KO^EGFR^) or with soluble CD109 (CD109-KO^CD109S^) alone (Figure 6D), suggesting that the loss of CD109 ablated membrane EGFR localization and raising the possibility that CD109 is required to maintain EGFR at the cell surface. Altogether, our results demonstrate that overexpressing EGFR, or GPI-anchored CD109 or soluble CD109 alone, cannot restore epithelial characteristics or signaling via the EGFR/AKT pathway; however, simultaneous overexpression of EGFR and GPI-anchored CD109 rescued the EGFR expression, AKT signaling, and the epithelial phenotype in CD109KO cells.

### 3.6. CD109 Is Required for EGF-Mediated Regulation of EGFR Levels, AKT Signaling, and to Maintain Stemness in SCC Cells

To assess the EGF-mediated regulation of EGFR levels, and AKT signaling, control A431 cells, CD109 KO, CD109KO^EGFR^, CD109KO^EGFR+CD109G^, and CD109KO ^EGFR+CD109s^ A431 cells were treated with 100 ng/mL EGF for 0 min, 5 min, 10 min, 30 min, 1 h, or 2 h. Following treatment, the cell lysates were analyzed by Western blot with specific antibodies for Phospho-EGFR, EGFR, Phospho-AKT, and AKT (Figure 7A). EGF stimulation caused a marked decrease in total EGFR levels at 120 min of EGF stimulation, as evident in control A431 and in CD109KO^EGFR+CD109G^ cells. We also observed that a substantial EGF-induced activation, followed by a decrease in phospho-EGFR and phospho-AKT, occurred in control A431 and CD109KO^EGFR+CD109G^ cells. As expected, the loss of CD109 completely diminished EGFR/AKT signaling. Furthermore, overexpression of EGFR alone or EGFR+soluble-CD109 was unable to rescue EGFR expression or its regulation by EGF to any substantial degree, or restore EGF-induced AKT signaling (Figure 7A). In contrast, CD109KO^EGFR+CD109G^ cells exhibited the same pattern of EGF-induced responses as control A431 cells (Figure 7A), suggesting that GPI-anchored CD109 is required for maximal propagation of EGF signaling and demonstrating that GPI-anchored CD109 drives SCC progression by sustaining the EGFR pathway.

To evaluate the role of CD109/EGFR in the maintenance of the cancer stem cell population, we examined the self-renewal property of control A431, CD109-KO, CD109KO^EGFR^, CD109-KO^CD109G^, CD109-KO^CD109S^, or CD109KO^EGFR+CD109G^ cells using an in vitro spheroid formation assay. Results shown in Figure 7B,C demonstrate that control A431 cells were able to grow as spheres, while CD109-KO cells showed a markedly decreased capacity to form spheroids under identical conditions. Forced expression of GPI-anchored CD109 alone, soluble CD109 alone, or EGFR alone in KO cells did not restore the spheroids’ formation ability, nor the forced simultaneous expression of EGFR and soluble CD109, in KO cells (Figure 7B). In contrast, forced simultaneous expression of both EGFR and GPI-anchored CD109 in CD109-KO cells restored their self-renewal ability, as evidenced by the capacity of CD109-KO^EGFR+CD109G^ cells to form tumor spheroids reaching comparable levels to that of control A431 cells (Figure 7B), suggesting that GPI-anchored CD109 is important for the maintenance of the stemness of SCC cells.

## 4. Discussion

Here, we identify CD109 as an essential regulator of EGFR at the mRNA and protein levels, and of EGFR/AKT signaling in vulvar and hypopharyngeal SCC cells, and of stemness and tumorigenicity in vulvar SCC cells. Furthermore, we show that the mechanism of CD109 action in SCC cells involves EGFR-CD109 heteromerization and colocalization, leading to the stabilization of EGFR levels. Additionally, we demonstrate that the maintenance of epithelial morphology and in vitro tumorigenicity of SCC cells require CD109 localization to the cell surface. Our study reveals that loss of CD109 markedly decreases the expression of stemness markers, and the tumorigenic potential of vulvar and hypopharyngeal SCC cells in vitro, and abrogates tumorigenicity and the metastatic ability of vulvar SCC cells in vivo. This, together with our findings above showing (i) that AKT pathway inhibition partially restores the phenotype of the CD109 KO SSC cells and (ii) that the simultaneous reintroduction of EGFR and CD109 expression (but not each one alone) rescues the phenotypic and functional features of the CD109 KO cells, indicates that CD109 exerts its pro-tumorigenic effects by associating with EGFR and promoting the EGFR/AKT activation.

While it is well documented that the expression of CD109 is dysregulated in SCC [14,16,17,36], whether it plays a causal role in SCC progression or whether it modulates key signaling pathways in SCC is poorly defined. In non-SCC cancers such as lung adenocarcinoma, several recent reports show that CD109 exhibits a pro-tumorigenic role, even though those reports vary as to the specific intracellular pathways that may mediate the CD109 effects. For example, the pathways reported potentially to mediate CD109′s tumorigenic effect in lung adenocarcinoma include JAK-SAT [19], Hippo-YAP [37], and EGFR-AKT-mTOR [38], but their relative significance and potential cross-talk in mediating CD109 action in lung adenocarcinoma are poorly understood. In comparison, there is only one study that has examined the role of CD109 in SCC and the regulation of signaling pathways. In that study, which examined the effect of CD109 in cervical SCC, it was reported that CD109 exhibits protumorigenic effects via the EGFR and STAT3 signaling [21]. However, the role of CD109 in other types of SCCs and the mechanism or involvement of other pathways in mediating the CD109 effect were not examined.

In the current study, our finding that loss of CD109 abrogates tumorigenicity of vulvar and hypopharyngeal SCC cells in vitro and of vulvar SCC cells in vivo, while strongly supporting the notion that CD109 plays a pro-tumorigenic role, was also somewhat unexpected. This is because our previous results showed that CD109 deletion in SCC cells enhances TGF-β signaling and promotes EMT traits in vitro by suppressing the epithelial phenotype and promoting SCC cell migration [18]. Although lack of consistency between in vitro 2D versus in vivo findings are not unusual, another possible explanation for this paradox (inhibition of EMT versus tumor promotion) is that the loss of CD109 leads to complete EMT, resulting in the loss of epithelial traits and the acquisition of a terminal mesenchymal phenotype [18]. Although it is generally held that cancer cells hijack the EMT process to enhance migratory, invasive, and stem-like properties and metastasis [39], emerging evidence suggests that it is the hybrid epithelial/mesenchymal (E/M) cells, rather than the cells that have undergone complete EMT, that are involved in cancer cell migration and invasion, resulting in metastasis [40]_._ Thus, it is possible that the loss of CD109 action leads to an irreversible and tumor-suppressive EMT program, as it generates a fully differentiated mesenchymal phenotype, resulting in the loss of tumorigenicity and metastatic ability [40,41]. Such a scenario provides a possible mechanistic explanation for how CD109 loss leads to EMT activation coupled with forfeiture of tumorigenicity in SCC cells. However, it is important to note that EMT is a reversible process regulated by multiple factors including master transcription factors and metabolic states [40,41], and thus the role of CD109 in the EMT process is likely to be context-dependent and may vary with in vitro versus in vivo conditions and 2D versus 3D cultures.

An important finding in the present study is that CD109 is required to sustain EGFR expression in vulvar and hypopharyngeal SCC cells (A431 and FaDu). Our findings reveal that GPI-anchored CD109 interacts and colocalizes with EGFR (Figure 4), and that this interaction is required to maintain EGFR expression in these SCC cells (Figure 4 and Figure 5). This, together with the observation that only simultaneous overexpression of EGFR and GPI-anchored CD109 (but not overexpression of EGFR alone, GPI-anchored CD109 alone, soluble CD109 alone, or simultaneous expression of soluble CD109 and EGFR) can restore cell surface EGFR expression in CD109 KO cells (Figure 6D), reverse the mesenchymal morphology (reverse the EMT state) of CD109 KO cells (Figure 6A), or rescue the CD109-EGFR colocalization (Figure 6D), and AKT activation (Figure 6B,C), establishes that CD109 is required for EGFR expression and function.

The current study provides insight into a key mechanism involved in SCC tumor formation. CD109 preferentially promotes the EGFR/AKT signaling and responses, as the levels of phospho-AKT are markedly decreased in all three CD109 knockout clones, while the levels of phospho-ERK and phospho-STAT3 were not altered in those clones (Figure 5A). Furthermore, in control SCC cells, the inhibitors of EGFR or AKT, but not inhibitors of TGF-β (Figure 5B,D) or ERK (data not shown), lead to a cell death response and notably to a mesenchymal morphology. Thus, AKT inhibition phenocopies inhibition of EGFR kinase, which also mimics the CD109 KO phenotype (Figure 5D). Together, these results demonstrate that the maintenance of epithelial cell morphology by CD109 involves EGFR/AKT signaling, and underscore the requirement of CD109 and the regulation of EGFR/AKT activation by CD109 in the maintenance of SCC cellular morphology.

Cancer stem cells (CSCs) are linked to tumor initiation, metastasis, and resistance to therapy in many cancer types, including SCC [42]. Understanding the regulation of stemness properties of SCC that define CSCs may allow for designing therapies that target specific functions of those cells for the treatment of SCC patients with poor prognoses. Our findings emphasize a critical role for CD109 in the maintenance of SCC cellular stemness. Notably, loss of CD09 diminishes the expression of markers of pluripotency/cancer cell stemness (SOX2, OCT4 and NANOG, and CD44) (Figure 3) and self-renewal/tumorigenic potential (spheroid formation) in vitro in SCC cells (A431, FaDu), demonstrating that CD109 is required for the maintenance of the cancer stem cell population in SCC. In line with our in vitro results, our in vivo findings highlight an essential role for CD109 in SCC tumor formation, as CD109-deficient SCC cells failed to form xenograft tumors, while control SCC cells formed tumors (Figure 1) and exhibited impaired formation of lung metastasis, while control SCC cells formed abundant lung metastatic lesions (Figure 2). Importantly, only the simultaneous overexpression of both EGFR and CD109 (but not EGFR alone, CD109 alone or soluble CD109 alone, or in combination with EGFR) is able to restore EGF-induced AKT signaling (Figure 7A) and the self-renewal ability of CD109-deleted SCC cells to levels comparable to that of control SCC cells (Figure 7B), demonstrating the requirement of CD109 for tumorigenicity of SCC cells. These findings are consistent with the recent reports that CD109 upregulation promotes stemness properties of non-SCC cancer cells such as lung adenocarcinoma cells [37] and that CD109 positive cancer cells exhibit higher tumorigenicity in epithelioid sarcoma [43], triple-negative breast cancer [44], and cervical squamous cell carcinoma [21].

In summary, our studies show that CD109 is required for subcutaneous tumor formation and lung colonization of vulvar SCC cells (Figure 8). Furthermore, our results demonstrate that CD109 plays an essential role in the maintenance of EGFR levels and that CD109 mediates EGFR/AKT activation in vulvar and hypopharyngeal SCC cells. In addition, we show that CD109 is required for the maintenance of SCC cellular morphology and its self-renewal ability. Together, these findings highlight a potential clinical utility for direct targeting of CD109 or selective inhibition EGFR/AKT signaling in SCC and warrant further validation using more clinically relevant orthotopic and patient-derived xenograft models.

## 5. Conclusions

Our findings reveal that CD109 is an essential regulator of EGFR expression at the mRNA and protein levels and of EGFR/AKT signaling in vulvar and hypopharyngeal SCC cells. Furthermore, we conclude that the mechanism of CD109 action in these SCC cells involves EGFR-CD109 heteromerization and colocalization, leading to the stabilization of EGFR levels, maintenance of epithelial morphology, stemness, and in vitro tumorigenicity. Additionally, our study identifies an essential role for CD109 in vulvar SCC progression in vivo.

## Figures and Tables

**Figure 1 cancers-14-03672-f001:**
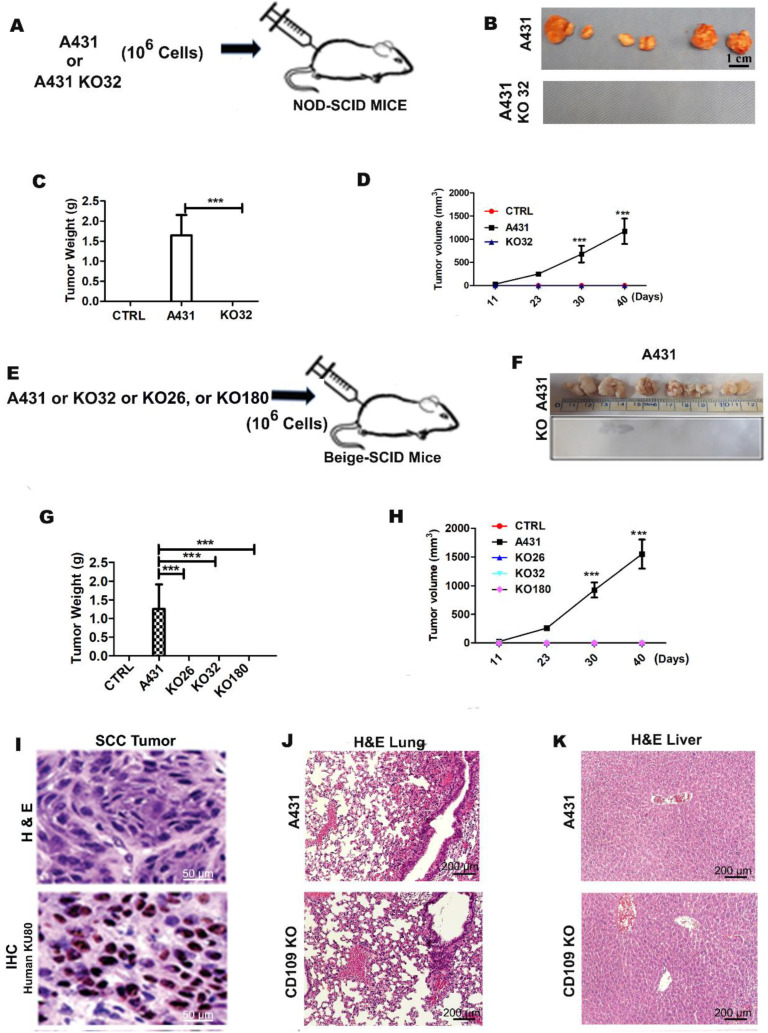
The loss of CD109 prevents tumorigenicity of A431 SCC cells in vivo. (**A**,**E**): A diagram of tumor Xenograft experiments. (**B**,**F**): The images of xenograft tumors from SCID mice injected with A431 (A431 CD109 wild type) or A431 CD109 KO cells as indicated. (**C**,**G**): Weight of xenograft tumors from SCID mice injected with A431 and CD109KO cells as indicated. (**D**,**H**): The growth (volume) curves of xenograft tumors derived from A431 or CD109KO cells as indicated. A 1:1 PBS/Matrigel (volume) was used as control. (**I**): Representative histology images of the lung tumor sections from control A431 cell tumors, as detected by hematoxylin-eosin (H&E) staining (Top). Sections were also subjected to immune-histochemical staining with anti-Human Ku80 (Bottom). (Note: Typical SCC has nests of squamous epithelial cells arising from the epidermis and extending into the dermis. The malignant SCC cells are often large with abundant eosinophilic cytoplasm and a large, often vesicular, nucleus. KO 32 clone represents A431 cells with CD109 deletion using CRISPR knock out strategy as described in methods). (**J**): Representative images of H&E histologic staining of lung sections. (**K**): Representative images of H&E histologic staining of liver sections. All the results are expressed as the mean ± S.D. Significance was calculated using a One-Way ANOVA *** *p* < 0.001.

**Figure 2 cancers-14-03672-f002:**
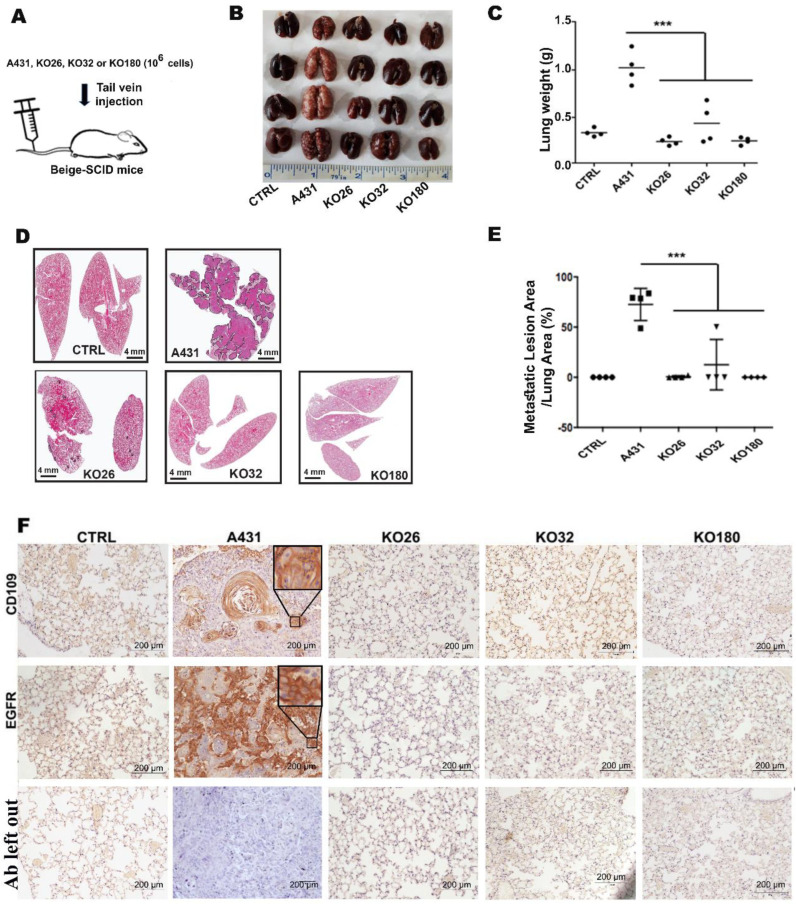
The loss of CD109 diminishes lung metastasis of SCC cells in vivo. (**A**): Experimental design of lung metastasis via tail vein assay. (**B**): Representative macroscopic images of lung metastases in formalin-fixed organs. (**C**): Quantification of lung weights in mice injected via tail vein with PBS control, A431 control cells, and three CD109 KO A431 cell lines as indicated. (**D**): Representative images of H&E-stained sections of lung tissues and lung metastases shown in B. (**E**): Quantification of lung tumor lesions in PBS control or A431 and three CD109 KO cells. All the results are expressed as the mean ± S.D. Significance was calculated using a One-Way ANOVA *** *p* < 0.001. (**F**): IHC staining for CD109 (top row), EGFR (middle row), and primary antibody left out control (bottom row) in the lung sections of mice injected with PBS (control), A431 control cells, KO180 A431, KO32 A431, or KO26 A431 cells.

**Figure 3 cancers-14-03672-f003:**
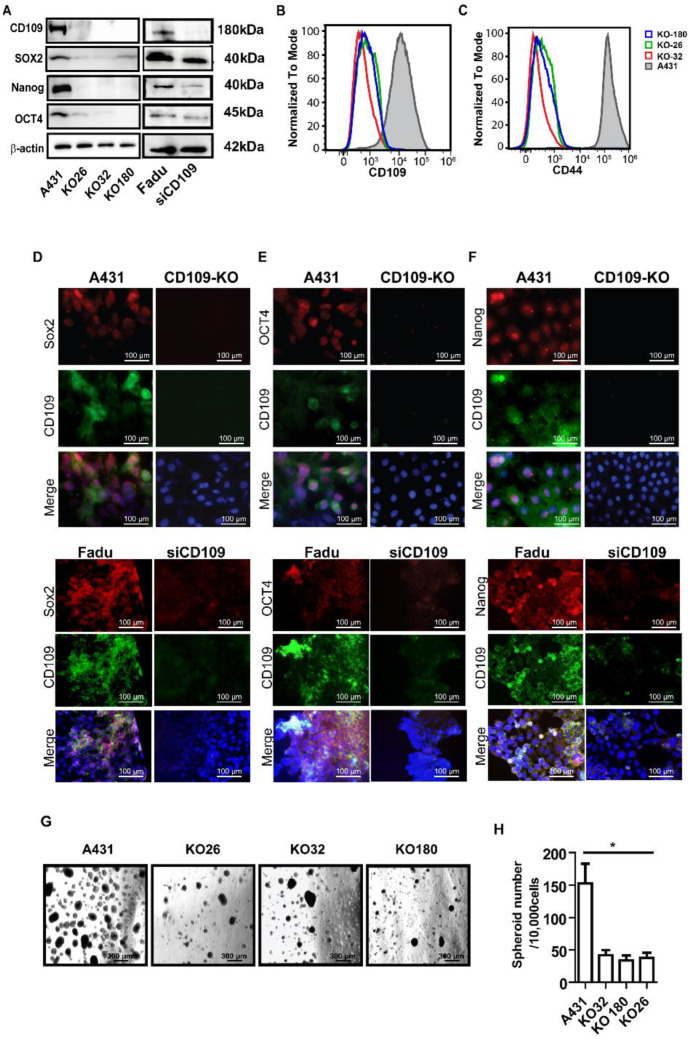
CD109 is required for maintaining the stemness of SCC cells, and the loss of CD109 diminishes cancer stem cell population in vitro in SCC cells. (**A**): Western blot analysis of OCT4, Sox2, and Nanog in control A431 versus CD109 KO A431, and control FaDu versus CD109-KD FaDu cells. (**B**,**C**): Representative images of Flow cytometry for CD109 (**B**) and CD44 (**C**), showing CD44 expression markedly diminished in CD109 KO cells. (**D**–**F**): Fluorescent microscopy of control A431 versus CD109 KO A431cells, and control FaDu versus CD109 KD cells: (**D**) Anti-Sox2 (red) and anti-CD109 (green); (**E**) Anti-Oct4 (red) and Anti-CD109 (green); (**F**) Anti-Nanog (red) and Anti-CD109 (green). (**G**): Representative phase-contrast images of spheroid formation assay with indicated cells. (**H**): Quantification of the number of tumor spheroids. All the results were expressed as the mean ± S.D. of three independent experiments. Significance was calculated using a One-Way ANOVA * *p* < 0.001. Uncropped WB images were shown in Appendix A.

**Figure 4 cancers-14-03672-f004:**
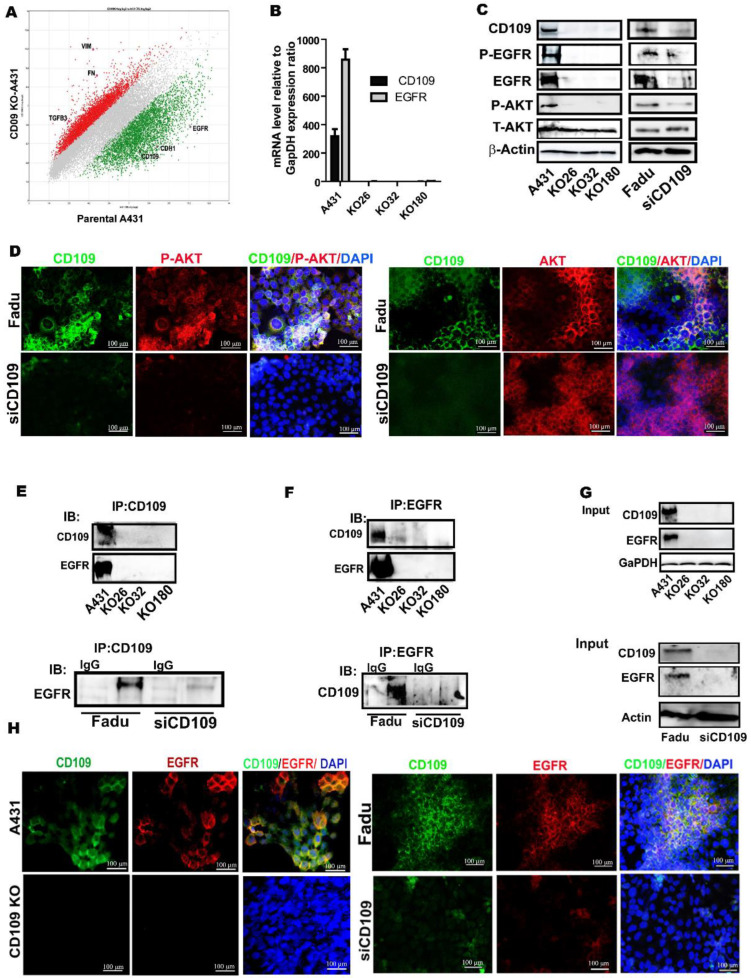
CD109 associates and co-localizes with EGFR, and loss of CD109 diminishes EGFR expression in SCC cells. (**A**): Scatter plot of microarray data from control A431 versus CD109KO A431 cells analysis indicates that EGFR is markedly downregulated in CD109 KO cells. (**B**): Quantitative PCR analysis of CD109 and EGFR gene expression in A431 control cells and three CD109 KO A431 clones. (**C**): Western blot analysis of indicated proteins in control A431 versus CD109KO A431 clones, and FaDu versus CD109-KD FaDu cells. (**D**): Representative images of immunofluorescence microscopy for FaDu and CD109-KD FaDu cells stained for CD109 (green), pAKT (red), and DAPI (blue). All the images are representative of three independent experiments. Scale bars: 100 μm. (**E**): Co-immunoprecipitation experiment where immunoprecipitation was conducted with an anti-CD109 antibody followed by western blot using anti-EGFR antibody, with control A431 versus CD109KO A431 clones, and FaDu versus CD109-KD FaDu cells. (**F**): Reverse co-IP where immunoprecipitation was conducted with an anti-EGFR antibody followed by western blot using anti-CD109 antibody, showing EGFR interacts with CD109. (**G**): Input control for the co-IP experiments. All the results were repeated in three independent experiments. (**H**): Immunofluorescence microscopy for control A431 versus CD109KO 32 A431 cells, and control FaDu versus CD109-KD FaDu cells stained for CD109 (green) and EGFR (red) and DAPI (blue). All the images are representative of three independent experiments. Scale bars: 100 μm. Uncropped WB images were shown in Appendix A.

**Figure 5 cancers-14-03672-f005:**
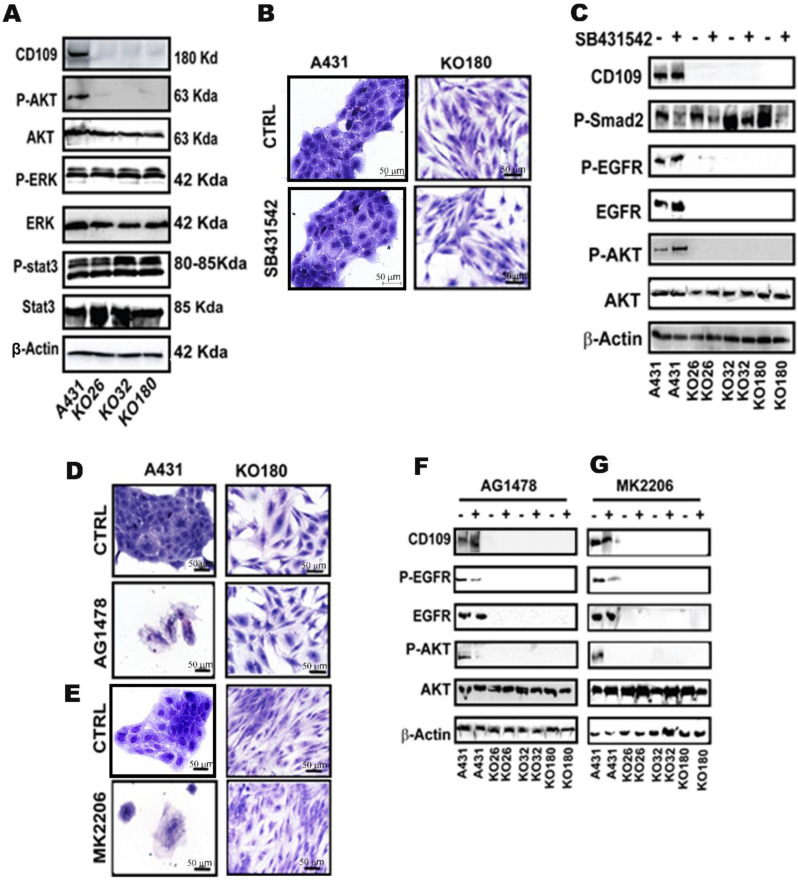
The AKT pathway, but not the ERK1/2 and STAT3 pathways, is inactivated in CD109 KO SCC cells, while in control SCC cells, EGFR kinase inhibition phenocopies AKT inhibition in inducing cell death and CD109KO-like mesenchymal state. (**A**): Western blot analysis for the indicated proteins in control A431 cells and CD109KO A431 clones. (**B**): Representative images of Toluidine blue staining for control A431 cells and CD109 109KO A431 treated with or without 10 μM TGF-β inhibitor SB431542 for 72 h. (**C**): Western blot analysis of cells in for the indicated proteins. (**D**,**E**): Representative images of Toluidine blue staining for control A431 cells and CD109KO A431 cells treated with or without as 5 μM/L EGFR inhibitor AG1478 (**D**) or AKT inhibitor MK2206 (**E**). (**F**,**G**): Western blot analysis of the control A431 and CD109KO A431 cells treated with or without 5 μM/L AG1478 (**F**) or 5 μM/L MK2206 (**G**) for 20 h. All the results were repeated in three independent experiments. Scale bar: 50 µm. Uncropped WB images were shown in Appendix A.

**Figure 6 cancers-14-03672-f006:**
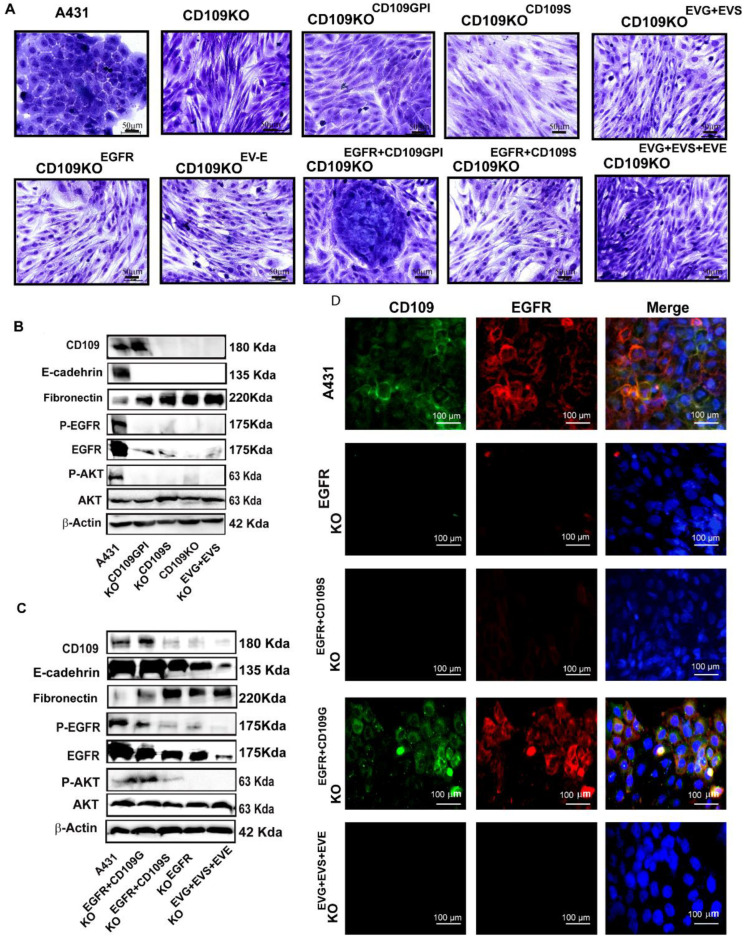
GPI-anchored CD109 is required for maintaining EGFR expression, levels of phospho-EGFR, and phospho-AKT, and the epithelial phenotype in CD109KO SCC cells. (**A**): Representative images of Toluidine blue staining of A431 and CD109-KO A431 cells transfected with indicated plasmids. Cell morphology was visualized by staining with 0.1% toluidine blue. (**B**,**C**): Western blotting analysis of cells in A for EGFR, phospho-EGFR, phospho-AKT, E-cadherin, and fibronectin. (**D**): Representative images of immunofluorescence microscopy for CD109 and EGFR using the cells indicated in (**A**). All results were repeated in three independent experiments. Scale bars: 50 µm or 100 μm as indicated. Uncropped WB images were shown in Appendix A.

**Figure 7 cancers-14-03672-f007:**
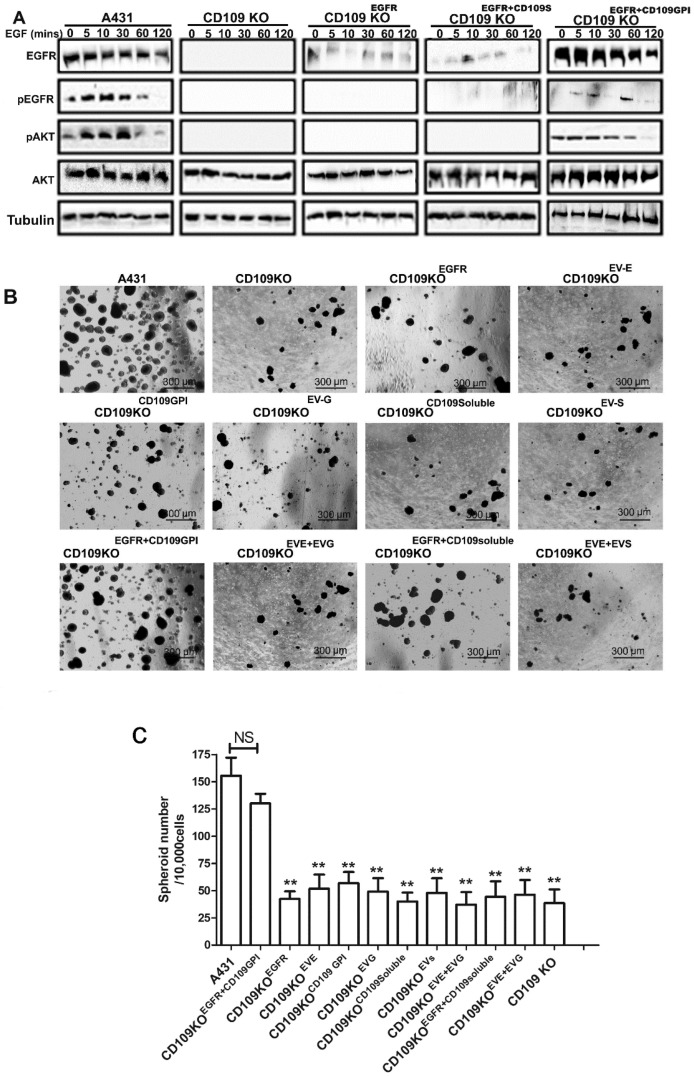
CD109 is required for EGF-mediated regulation of EGFR levels, AKT signaling, and to maintain stemness in SCC cells. Co-expression of GPI-anchored CD109 and EGFR (but not EGFR or GPI-anchored CD109 or soluble CD109 alone or co-expression of soluble CD109 with EGFR) restores EGFR expression, phospho-EGFR, and phospho-AKT levels, and tumorigenicity in CD109 KO SCC cells. (**A**) Western blot analysis for the indicated proteins of control A431 cells or CD109KO cells transfected with the indicated plasmids. All cells were serum-starved overnight, then treated with 100 ng/mL of EGF for indicated time points before Western blot analysis. (**B**) Representative images of tumor spheroids formed by control A431 cells or CD109KO cells transfected with the indicated plasmids as in A. (**C**) Quantification of the number of tumor spheroids formed in B. Five random fields were photographed, and the number of tumor spheroids was analyzed. All the results were expressed as the mean ± S.D. of three independent experiments. Significance was calculated using a One-Way ANOVA: ** *p* < 0.01. Uncropped WB images were shown in Appendix A.

**Figure 8 cancers-14-03672-f008:**
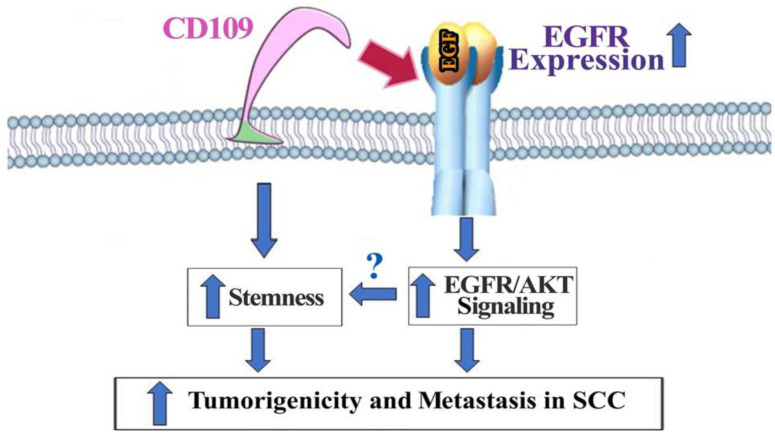
Schematic representation of the proposed mechanisms by which CD109 acts as an essential regulator of EGFR expression and EGFR/AKT signaling and a driver of tumor formation in squamous cell carcinoma. CD109 interacts with EGFR to maintain the levels of EGFR expression, promote EGFR/AKT signaling, and increase cancer cell stemness via EGFR/AKT or other pathways to increase tumorigenicity and metastasis in SCC.

**Table 1 cancers-14-03672-t001:** Primer sets used for qPCR.

human GAPDH:	forward primer	5′-GACAACTTTGGTATCGTGGAAGG-3′
	reverse primer	5′-AGGGATGATGTTCTGGAGAGCC-3′
human EGFR:	forward primer	5′-AAACCGGACTGAAGGAGCTG-3′
	reverse primer	5′-CCCATTGGGACAGCTTGGAT-3′
human CD109:	forward primer	5′-CTGGAACACTGCCCTTCACA-3′
	reverse primer	5′-GTCCGGTTACACGTAGCTCA-3′

## Data Availability

All data are available in the text and Appendix A.

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
