# Peer review of "CD109 Is a Critical Determinant of EGFR Expression and Signaling, and Tumorigenicity in Squamous Cell Carcinoma Cells"

_cancers, 2022, doi:10.3390/cancers14153672_

Round 1

Reviewer 1 Report

This study investigated the role of CD109 in tumour formation, lung metastasis, the maintenance of epithelial morphology and stemness, and the EGFR/AKT signalling and its associated protein complex formation, distribution and stabilisation in vulvar and hypopharyngeal SCC cell lines. Although the study is interesting, there are some major concerns about the quality of data presentation in some figures, the lack of enough controls and references in places, and also over-interpretation. The proofreading is absolutely needed as well.  

Major

The study suffers from a major drawback of not enough A431 controls as only one clone was used along with three KO clones in many experiments. Ideally, at least two control clones or the mixed clones or parental A431 cells should be included in the study. The authors tried to back up by using the FaDu cells with RNAi, but all the animal work was based on A431 cell lines. Alternatively, the authors can validate the A431 control clone by a knockdown study and demonstrate the phenotype is indeed CD109 specific by using various in vitro assays.

Another major concern is that the quality of many blots and IMF images in the figures is poor, e.g. Figure 3D~F, Figure 4D&H, Figure 6D and EGFR/pEGFR blots in various figures.

In addition, the references are missing in places, e.g. the CD109GPI and CD109S constructs, Pg9 line 11, Pg15 lines 19~24, etc.  

Pg7: confusion about the stable/transient transfection of EGFR and CD109.

Fig. 1I: What are the typical SCC traits? Why no CD109 staining? Is KO32 not the A431 cells? Need to label them correctly. The info about the p-value and scale bars is missing in the figure.

Fig. 2: No p values at * and ** in the figure and no scale bars too.

Fig. 3A: Ideally, to include A431-P alongside the Ct clone and to show that the Ct clone of A431 is truly representative.

Pg10 line 27 “We observed that most of the KO cells quickly underwent apoptosis, …” No such apoptosis data was shown. Over interpretation.

Fig. 4: The quality of EGFR/pEGFR in FaDu cells is poor. Also, it is not a good study design to use the CD109 KO line to evaluate the protein complex formation between CD109 and EGFR. It is not surprising to see the results as CD109 was lost in the cells. Thus, the comment at Pg11 line 38 “… but not in CD109-KO and CD109-KD FaDu cells.” is meaningless.   

Figure 4H: A431, which KO clone cells were displayed?

Fig. 5: Please check through all the symbols that are correctly shown, e.g. page 12 line 10: 10?M/mL; line 14 (????). In fact, there are many errors throughout the MS, and these must be corrected. 

Figure 6:

Pg13 line 12~13: what’s the EMT marker? No reference(s) is shown. Quantitation for the blots of the EMT marker in Figure 6B is needed. Line 17 on the same page: The transient transfection of EGFR was missing in M+M or is this correctly described.

Why no detection was shown for the soluble CD109 expression by Western blotting analysis? The authors need to validate the transfection of CD109S was working. 

Fig. 6D the quality of images is poor. Ideally, the restoration of surface EGFR in KO-EGFR+CD109G cells needs to be confirmed by the surface biotinylated assay. 

Fig. 7: again, poor quality in some blots. Pg14 line 3~7: Need the quantification.

Pg15 lines 42~44 Discussion: “It is possible that CD109 action leads to an irreversible tumor-suppressivesive EMT program, as it generates fully differentiated mesenchymal phenotype, resulting in the loss of tumorigenicity and metastatic ability [42].” This statement is contrary to the findings from this study as well as to the model shown in Figure 8.

Pg4 line 31: “2.4. In vivo animal studies” The section numbering is wrong

Proofreading is absolutely necessary as there are several typos and space issues.

Pg6 line 36: 48C?

Minor

The country is missing in the affiliations

Pg3 line 43: the age of the patient from whom the FaDu cell line was derived is missing.

Pg5 line 20: the volume of PBS?

Reviewer 2 Report

The manuscript ”CD109 Is A Critical Determinant of EGFR Expression and Signaling, And Tumorigenicity In Squamous Cell Carcinoma Cells” by Shufeng Zhou investigates the role of CD109 in stemness, epithelial morphology, AKT signaling and stabilization of EGFR levels in SCC cells via CD109-EGFR interaction.

The paper is clearly written and the study subject is interesting and to some extent novel for the chosen cancer type. The majority of the claims is plausible and supported by the experimental data however the experimental data presented is in some cases of poor quality. 

For publication, several aspects need to be addressed:

Growth rate /cell doubling time / viability

The authors use A431 cells and A431 derived cd109KO clones for their in vitro and in vivo analysis. Several of the assays are dependent on the growth rate of the cell lines (for example spheroid assays). Are there differences in proliferation between the parental cell line and the KO variants? If, so many of the differences could be due to growth delay rather than stemness, migration potential etc. 

Before injecting A431 cells and clones into mice, did the authors perform viability testing to ensure that all cells where in good condition?

Western blotting

Did the authors stain total protein and phosphorylated proteins on the same membrane? If not, please provide loading controls for all corresponding protein bands. 

Please provide clear images of the western blot membranes. For example in Figure 4 C. The membrane staining is very confusing. Is there a higher expression of p-EGFR than total EGFR in the Fadu siCD109 cells?

The authors describe significant difference of protein expression measured by western blotting several times in the manuscript. What significance test was used? Did they perform any quantification? Please provide graphs displaying the quantification of the protein expression. 

Confocal microscopy

The images are generally very small with low resolution. The shown expression is within representative images is heterogenous, for example Figure 3. D, E, F. Here the CD109 signal seems very heterogeneous. Is that due to different expression of the protein on different cells or staining artefacts?

Figure 4 H. It is not optimal that the size references takes up one-sixth of the entire image.

-Figure 3. D, E, F. Please indicate what nuclear stain was used. 
For all experiments show images of the nuclear stain alone without green and red channels. 

Minor concerns:

- Please check all special characters in the manuscript. They are displayed wrongly in the entire manuscript. For example Greek letters as in TGF-beta.

- Figure 1 B. Please add size reference to indicate dimension of the dissected tumors.

- Figure 1 D and H. It is impossible at the current resolution to tell the difference between the groups. Please use suitable resolution and/or use colors or larger symbols.

-Figure 2 E: Figure legend “Quantification of lung weights in mice injected via tail vein with PBS control, A431 control cells or one of the three CD109 KO A431 cell lines as indicated” Should be “and three CD109 KO A431 cell lines”.

-Figure 2 F. Please mention in the legend what the third row of IHC indicates. Is that only the counterstain? If so, what are the brown areas?

- Figure 3 B and C. The x-axis lacks a scale. Is it linear or exponential?

Author Response

Please the attachment.

Round 2

Reviewer 1 Report

Thanks to the authors for addressing the concerns/issues raised in the first round of review which is greatly appreciated. Although the MS has been improved, unfortunately, there are still several issues that remain outstanding and the justification is required in many places throughout the MS.

Proofreading issue: please go through the MS carefully, e.g. abstract line 29 should be ‘were’; space issues such as pg3 line 2 and 6, line 21, pg10 line 27, and many more; pg5 line 33 ‘on lung’; pg6 line 39. ‘were’, pg6 line 25 ‘The proteins were extracted from the whole cell lysates.’ (doesn't make sense), Pg6 line 3: ‘For FaDu cells, Co-IP 3 was done using Protein G magnetic beads as per the manufacturer’s protocol (Biorad; 1614023) ….’,

Pg4 line 27: why cells were starved, prior to the analyses? Please justify

2.10: the description for western blotting is incomplete

Pg10 line 9: Is this a conclusion based on the first part of the study ‘To better understand the mechanism by which CD109 may exert its pro-tumorigenic effect of SCC cells in vivo, …’

Pg11 line 5: ‘… the loss of CD109 significantly diminishes the cell population in SCC cells.’ Did the authors tried to say that the loss of CD109 significantly diminishes the spheroid forming capacity in SCC cells as this experiment didn’t measure the cell number rather than spheroids?

Line 45: ‘ …confirming that CD109 interacts with EGFR in control A431 and FaDu cells’. IMF does prove the protein interaction.

Pg12 line 21: Is this correct 10μM/mL and also line 26?

Pg14 line 22: phosphor-AKT, please correct

Pg15 line 4: Here we identify CD109 as an essential regulator of EGFR levels at the mRNA and protein levels, …

Line 12: Our study reveals that loss of CD109 markedly expression of stemness markers, … please also double-check the next sentence.

For the p values in the figure and figure legends, it’s better to standardise the p values throughout the MS rather than define them individually in each figure.

Please see below my responses to some of the points (file attached)

Round 3

Reviewer 1 Report

please include in the legends only the p values shown in the figures.
